# Cyclophilin BcCyp2 Regulates Infection-Related Development to Facilitate Virulence of the Gray Mold Fungus *Botrytis cinerea*

**DOI:** 10.3390/ijms22041694

**Published:** 2021-02-08

**Authors:** Jiao Sun, Chen-Hao Sun, Hao-Wu Chang, Song Yang, Yue Liu, Ming-Zhe Zhang, Jie Hou, Hao Zhang, Gui-Hua Li, Qing-Ming Qin

**Affiliations:** 1College of Plant Sciences, Key Laboratory of Zoonosis Research, Ministry of Education, Jilin University, Changchun 130062, China; sunjiao18@mails.jlu.edu.cn (J.S.); liuyue_19@mails.jlu.edu.cn (Y.L.); nihaojiehou@163.com (J.H.); 2College of Plant Sciences, Jilin University, Changchun 130062, China; sunch16@mails.jlu.edu.cn (C.-H.S.); songyang19@mails.jlu.edu.cn (S.Y.); mzzhang@jlu.edu.cn (M.-Z.Z.); 3Key Laboratory of Symbolic Computation and Knowledge Engineering, Ministry of Education, College of Computer Science and Technology, Jilin University, Changchun 130012, China; changhw17@126.com (H.-W.C.); zhangh@jlu.edu.cn (H.Z.); 4College of Forestry, Beihua University, Jilin 132013, China

**Keywords:** *Botrytis cinerea*, cyclophilin, calcineurin, morphogenesis, infection structure formation, pathogenesis, cyclosporine A (CsA)

## Abstract

Cyclophilin (Cyp) and Ca^2+^/calcineurin proteins are cellular components related to fungal morphogenesis and virulence; however, their roles in mediating the pathogenesis of *Botrytis cinerea*, the causative agent of gray mold on over 1000 plant species, remain largely unexplored. Here, we show that disruption of cyclophilin gene *BcCYP2* did not impair the pathogen mycelial growth, osmotic and oxidative stress adaptation as well as cell wall integrity, but delayed conidial germination and germling development, altered conidial and sclerotial morphology, reduced infection cushion (IC) formation, sclerotial production and virulence. Exogenous cyclic adenosine monophosphate (cAMP) rescued the deficiency of IC formation of the ∆*Bccyp2* mutants, and exogenous cyclosporine A (CsA), an inhibitor targeting cyclophilins, altered hyphal morphology and prevented host-cell penetration in the *BcCYP2* harboring strains. Moreover, calcineurin-dependent (CND) genes are differentially expressed in strains losing *BcCYP2* in the presence of CsA, suggesting that BcCyp2 functions in the upstream of cAMP- and Ca^2+^/calcineurin-dependent signaling pathways. Interestingly, during IC formation, expression of *BcCYP2* is downregulated in a mutant losing *BcJAR1*, a gene encoding histone 3 lysine 4 (H3K4) demethylase that regulates fungal development and pathogenesis, in *B. cinerea*, implying that BcCyp2 functions under the control of BcJar1. Collectively, our findings provide new insights into cyclophilins mediating the pathogenesis of *B. cinerea* and potential targets for drug intervention for fungal diseases.

## 1. Introduction

*Botrytis cinerea* is a typical necrotrophic fungal pathogen that causes gray mold on over 1000 plant species, including almost all vegetable and fruit crops [1], and annually causes economic losses of USD 10 to 100 billion worldwide [2]. Gray mold of grape, tomato and strawberry is commonly seen. The pathogen mainly infects the remaining stigmas or petals, and then slowly spreads to fruits, causing grayish-white and later a thick layer of gray mold on the infected fruits. The symptoms of leaves often start from leaf-tips, and the lesions spread in a “V” shape along the veins and appear water-soaked [1,3]. 

In natural conditions, the main infection source of the pathogen is conidia that attach on the surface of plants, germinate and form appressoria or appressorium-like infection structures to facilitate host penetration [4]. Mycelia of the pathogen can form infection cushions (ICs), another kind of infection structure with a special morphology of highly melanized “specialized hyphal networks” or “clumps of hyphae”, on the plant surfaces to invade host cells/tissues [5,6]. Besides appressoria and ICs, germ tube apices of the pathogen also occasionally directly penetrate host cells [7]. During interactions with its hosts, the pathogen employs diverse virulence-associated factors, including cell wall degrading enzymes, cutinases, toxins, hormones, small RNAs and other pathogenesis-related factors, to facilitate host infection. These factors function coordinately to enable the pathogen to induce the silencing of host immune response genes, kill host cells, break down the dead host tissues and assimilate nutrients from the killed host cells to support its growth [2,8,9]. When the nutrient of hosts is exhausted, the pathogen forms sclerotia to survive in unfavorable environments for long periods. When the environment is suitable, the sclerotia germinate, form mycelia and conidiophores, and produce a large number of conidia; with the aids of wind, rain, irrigation water and farming tools, conidia land on plant surfaces and start a new round of infection [3].

Cyclophilins (Cyps) are a conserved family of proteins named after their ability of binding to cyclosporin A (CsA), an immunosuppressant used to suppress rejection after internal organ transplants [10,11,12]. The cytosolic cyclophilin A is an intracellular receptor of CsA and mediates CsA function in immunosuppression [13]. Cyclophilin has the activity of peptidyl-prolyl cis-trans isomerase (PPIase), a rate-limiting enzyme that plays an important role in protein folding and assembly [14,15,16]. During the interaction of cyclophilin and CsA, CsA binds to cyclophilin via contact within the hydrophobic pockets and inhibits the PPIase activity [11]. The resulting cyclophilin/CsA complex is a strong inhibitor of calcineurin (CN, also known as protein phosphatase 2B) [17,18]. Calcineurin is a Ca^2+^/calmodulin-dependent serine/threonine protein phosphatase and a crucial mediator of intracellular signaling that couples Ca^2+^ signals to many cellular responses or physiological processes, including hyphal branching and morphology, formation of infection structures, sexual development, environmental stress adaptation (e.g., alkaline pH, high concentrations of NaCl, CaCl_2_, and MgCl_2_), and virulence in pathogenic fungi [17,18,19,20,21]. 

Cyclophilins are involved in a variety of cellular processes, including response to environmental stresses, regulation of cell cycle, calcium signaling, and transcriptional repression [22]. The processes are also associated with human pathologies such as neurodegenerative diseases, infectious diseases and cancer. Therefore, cyclophilins are expected to become new targets for the cure of these diseases [23,24,25,26,27]. In pathogenic fungi, cyclophilins play a crucial role in vegetative growth and virulence. *Beauveria bassiana*, a filamentous insect-pathogenic fungus, contains 11 cyclophilins, among them BbCypA has the highest expression level during growth and shows a sensitive PPIase activity when expressed in vitro. Except for the *BbCYPA* deletion mutant, strains with deletion or overexpression of any other *CYPs* exhibit a certain degree of temperature-sensitive phenotype. Loss of *BbCYPE* and *BbCYP6* impairs the virulence of the mutant strains [28]. MoCyp1 in the rice blast fungus *Magnaporthe oryzae* is a virulence-associated factor that mediates appressorium development and virulence by regulating Ca^2+^ signaling [29]. In *B. cinerea*, loss of *BCP1*, a gene that encodes a cyclophilin protein Bcp1, reduces virulence of the mutant strain [17]. *Cryphonectria parasitica*, the chestnut blight fungus, relies on cyclophilin Cyp1 to sense signals in the environment, and this protein plays an important role in the virulence of the pathogen [30]. 

The genome of *B. cinerea* contains 16 putative cyclophilins (Appendix A). Although the cyclophilin A protein Bcp1 encoded by *BCP1* has been characterized [17], the functions of the remaining cyclophilins in the mediation of the fungus development and pathogenesis remain unexplored. In this study, using a gene disruption strategy, we demonstrate that a putative cyclophilin protein, BcCyp2, is a novel virulence-associated factor. We found that the *BcCYP2* deletion mutant ∆*Bccyp2* was less sensitive to CsA, suggesting that BcCyp2 functions as a cyclophilin protein and is the target of CsA. Disruption of *BcCYP2* impairs the pathogen IC formation and virulence. Furthermore, exogenous cAMP (cyclic adenosine monophosphate) rescues the defect in the ability to form ICs of the mutants. Our findings indicate that cyclophilin BcCyp2 is a virulence-associated factor of the gray mold fungus and provide new insights into the pathogenesis of *B. cinerea*. 

## 2. Results 

### 2.1. BcCYP2 Is a Virulence-Associated Gene in B. cinerea

To identify virulence-associated factors that mediate pathogenesis of *B. cinerea*, and to understand molecular mechanisms underlying the interactions between *B. cinerea* and its hosts, we screened pathogenicity-attenuated mutants from a *B. cinerea* library that contains ~50,000 transformants generated by *Agrobacterium tumefaciens*-mediated transformation (ATMT) approach [31,32,33] and identified pathogenicity-reduced (on both detached tomato and strawberry leaves) mutant strain M331. Thermal asymmetric interlaced PCR (TAIL-PCR) and sequencing analysis of the strain indicated that a T-DNA was inserted into the position of 800 bp coding region of an ORF (Appendix A) that has been previously annotated as a gene encoding a hypothetical protein BCIN_05g03080 [34]. Bioinformatics analysis suggests that the hypothetical protein is a cyclophilin_ABH_like protein (Appendix A) and shares high homology with the *B. cinerea* Cyp protein Bcp1 [17]. Therefore, the T-DNA tagged gene was designated as *BcCYP2.*

To investigate the roles of *BcCYP2* in the pathogen growth and virulence, we generated *B. cinerea BcCYP2* knockout (KO) mutant ∆*Bccyp2* with the illustrated strategy (Appendix A) and its complemented strain ∆*Bccyp2*-C as previously described [6,35,36]. The deletion of *BcCYP2* in the mutant strains was confirmed by PCR detection and quantitative reverse transcription PCR (qRT-PCR) analysis (Appendix A). We then performed pathogenicity assays for the wild-type (WT), the T-DNA tagged mutant M331, ∆*Bccyp2*, and the complemented ∆*Bccyp2*-C strains using conidium–inoculation approach. Consistent with the attenuated pathogenicity in the T-DNA tagged mutant strain M331, loss of *BcCYP2* significantly reduced virulence of the ∆*Bccyp2* mutants. Complementation of the mutant strain ∆*Bccyp2*-1 with the *B. cinerea* WT *BcCYP2* locus rescued the pathogenicity defect of the mutant (Appendix A). The results indicate that *BcCYP2* is a virulence-associated gene in *B. cinerea*.

### 2.2. BcCYP2 Mediates B. cinerea Conidial Development and Morphogenesis but Is Dispensable for Conidiation and Radial Growth of Mycelia

To test whether *BcCYP2* regulates the growth of *B. cinerea*, we determined the mycelial growth rate of the WT, ∆*Bccyp2* and ∆*Bccyp2*-C strains growing on complete medium (CM) plates. Our results demonstrated that all the tested strains failed to display a significant difference in mycelial radial growth during a time course of 3 (for mycelial plugs) or 4 (for conidia) days of incubation (Appendix A). However, conidial development assay in a time course of 6 h indicated that loss of *BcCYP2* in *B. cinerea* delayed conidial germination and germling development (Appendix A). Together, these data indicate that *BcCYP2* is required for *B. cinerea* conidial germination and germling development but is dispensable for radial growth of mycelia.

To investigate whether *BcCYP2* plays a role in asexual reproduction and conidial morphogenesis, we inoculated conidial suspensions of the test strains on CM plates, and then determined their sporulation ability and conidial morphology at 10 days post-incubation/inoculation (dpi). Our data indicated that all the tested strains displayed similar conidiation abilities (Figure 1a,b). The morphological analysis of conidia produced by the tested strains showed that the conidial length of the ∆*Bccyp2* mutant (8.81 ± 1.08 μm) was significantly shorter than that of the WT and complemented conidia (11.5 ± 1.11 μm and 11.39 ± 1.18 μm, respectively). However, the conidial width of all tested strains (7.05 ± 1.35 μm, 7.06 ± 1.42 μm, and 7.05 ± 1.36 μm for the WT, ∆*Bccyp2* and complemented strains, respectively) was similar. The length/width ratio of the mutant conidia was 1.22 (1.63, 1.61 for the WT and complemented conidia, respectively), which made the ∆*Bccyp2* mutant conidia smaller, more globose and less elliptical than controls (Figure 1c–e). These data demonstrate that *BcCYP2* is dispensable for *B. cinerea* conidiation and mycelial growth but plays an important role in conidial morphogenesis.

### 2.3. BcCYP2 Is Dispensable for B. cinerea Cell Wall Integrity and Stress Adaptation

To test whether loss of *BcCYP2* affects *B. cinerea* adaptation to infection-related stresses, we compared the radial growth rates of the WT, Δ*Bccyp2*, and complemented strains on CM containing the osmotic stress agents NaCl and KCl, the oxidative stress agent H_2_O_2_, or the cell wall disturbing agents sodium dodecyl sulfate (SDS) and Congo red [6,31,32,36]. Our results showed that all the tested strains, no matter inoculated with conidia (Appendix A) or mycelia (Appendix A), displayed similar mycelial growth rates on CM supplemented with the indicated stress-mimic agents (Appendix A). These data suggest that *BcCYP2* is dispensable for the pathogen mycelial osmotic- and oxidative-stress adaptation as well as cell wall integrity.

### 2.4. BcCYP2 Regulates B. cinerea Sclerotium Production and Morphogenesis

To determine the role of *BcCYP2* in sclerotium production, we inoculated conidia of the WT, Δ*Bccyp*2, and Δ*Bccyp2*-C on CM plates, incubated these plates at 20 °C in dark condition for four weeks and analyzed sclerotium formation. We found that the number of sclerotia formed by the Δ*Bccyp2* mutant strains was only about 35% of that of the WT or Δ*Bccyp2*-C strains (Figure 2a,b). However, the size of the mutant sclerotia was bigger than that of sclerotia produced by the WT or complemented strains (Figure 2a,c). Sclerotial germination assays indicated that germination rates of the normal and abnormal sclerotia produced by the WT or Δ*Bccyp2*-C, and Δ*Bccyp2* strains, respectively, were not significantly different (Figure 2d). These findings suggest that *BcCYP2* mediates *B. cinerea* sclerotium production and morphogenesis but is dispensable for sclerotial germination.

### 2.5. BcCYP2 Regulates IC Formation but Is Dispensable for Appressorium Formation 

To evaluate the effect of *BcCYP2* on infection structure formation, we inoculated the WT, Δ*Bccyp2*, and Δ*Bccyp2*-C strains on glass slides or PVC (Polyvinyl chloride) sheets and determined the formation of appressoria and ICs. Our results indicated that all the tested strains produced similar numbers of appressoria on the glass slides or PVC sheets at 7 h post-inoculation/incubation (hpi) (Figure 3a,c,d), suggesting that *BcCYP2* is dispensable for the pathogen appressorium formation.

The initial formation of ICs was observed at ~16 hpi in the WT and complemented strains; however, for Δ*Bccyp2* mutants, the ICs were firstly observed until 20 hpi (Figure 3b). In addition, the ICs produced by the mutants were significantly smaller in size than those of the WT and complemented strains at 24, 36 or 48 hpi (Figure 3b,e). The results indicate that *BcCYP2* mediates the development of ICs in *B. cinerea*.

### 2.6. BcCYP2 Is Required for Full Virulence of B. cinerea

To investigate the role of *BcCYP2* in *B. cinerea* virulence, we inoculated green bean leaves with conidial suspensions (2 × 10^5^ conidia/mL, 5 μL) of the WT, Δ*Bccyp2*, and Δ*Bccyp2*-C strains and determined their pathogenicity. Our result showed that the Δ*Bccyp2* mutants could induce lesions on the host leaves; however, the lesion size induced by the mutant strains was smaller than that induced by the WT or complemented strain (Figure 4a,f). The results indicate that *BcCYP2* is required for *B. cinerea* full virulence.

To further analyze the role of *BcCYP2* in host penetration, we inoculated onion epidermal cells with conidia of the tested *B. cinerea* strains and observed the infection and development of the pathogen. The results indicate that the mutant has the ability of host penetration, but its penetration rate was lower than that of the WT or complemented strains at 12 hpi and the penetration rates were similar at 24 hpi (Figure 4b,g). The results indicate that *BcCYP2* plays a role in the initial host penetration of *B. cinerea*.

To determine whether *BcCYP2* is involved in the pathogen expansion in host tissue upon penetration, we inoculated host leaves or apple fruits with mycelial plugs of the tested strains by a non-wound- or wound-inoculation method, the latter allows the pathogen to enter host cells through an infection structure-independent manner. Our findings indicated that the lesion sizes induced by all the tested strains failed to display a significant difference in the wound-inoculation assay (Figure 4c–e,h–j), suggesting that *BcCYP2* plays a limited role, if any, in the pathogen expansion in planta. However, in the non-wound-inoculation, the mutant strains displayed a reduction in lesion expansion compared to the WT or complemented controls (Appendix A). Taken together, these data suggest that *BcCYP2* is required for the initiation of host penetration but plays a limited role in invasive hyphal expansion in planta.

### 2.7. Exogenous cAMP Restores the Ability of BcCyp2-Deficient Mutants to Form ICs

cAMP can promote the formation of appressoria and ICs in phytopathogenic fungi, including *B. cinerea* [6,36]. To determine whether a delay in IC formation in the Δ*Bccyp2* mutants is regulated by cAMP signaling, we inoculated conidia (1 × 10^5^ conidia/mL, 10 µL) of the tested strains supplemented with exogenous cAMP (0, 20, 50, 80 μM) on glass and determined infection structure formation at the indicated hpi. Our findings demonstrated that exogenous cAMP restored the ability of IC formation of the Δ*Bccyp2* mutants to that of the WT or complemented strain without cAMP treatment (Figure 5). The result demonstrated that cAMP could rescue the defect of infection-related morphogenesis of the mutants losing *BcCYP2*, and suggests that BcCyp2 likely functions upstream of cAMP signaling.

### 2.8. CsA Suppresses Conidial Germination, Hyphal Morphogenesis and Host Penetration of B. cinerea Strains Harboring CYP2

To determine whether BcCyp2 is a target of CsA, an immunosuppressive agent that binds to the cyclophilin and then inhibits calcineurin [11,17], we tested the sensitivity of the mutant strain Δ*Bccyp2* to CsA. Our results demonstrated that conidial germination and hyphal development of the WT strain were dramatically suppressed. Importantly, hyphal branching of the WT strain was also severely altered (Figure 6a; Appendix A). Compared to the WT strain, conidial germination and hyphal development of the Δ*Bccyp2* strain displayed less inhibition by low concentration of CsA (0.2~0.4 μg/mL). The suppression of conidial germination showed a CsA concentration-dependent fashion, and the inhibition phenotype of the mutant tended to be the same as the WT strain when higher doses of CsA were used (Figure 6a; Appendix A). The results suggest that BcCyp2, as a functional cyclophilin protein, is one of the targets of CsA.

To test the effect of CsA on fungal infection-related development and host-cell penetration, we then inoculated onion epidermal cells with conidial droplets (containing 0.4 μg/mL CsA) of the tested strains and then analyzed the effects of CsA on Δ*Bccyp2* infection. Our results showed that at 24 hpi, the CsA-treated WT and complemented strains failed to invade the onion epidermis; however, penetration of the onion epidermis by the CsA-treated mutant strains was not affected (Figure 6b). At 48 hpi, host penetration by the WT and Δ*Bccyp2*-C strains was still hardly detected (Figure 6b). These findings indicate that suppression of fungal infection-related development and host-cell invasion by the immunosuppressant CsA is in a BcCyp2-dependant fashion; and imply that CsA functions via interaction with BcCyp2 which results in blocking host invasion by the pathogen.

### 2.9. BcCyp2 Mediates CsA-Regulation of Calcineurin Dependent (CND) Genes

The less hyphal morphological alteration and the unaffected infection structure formation in strains lacking *BcCYP2* in the presence of CsA suggest that *BcCYP2* may be associated with Ca^2+^/calmodulin-dependent phosphatase calcineurin. To test this possibility, we cultured the WT, mutant, and complemented strains with media supplemented with 10 µg/mL CsA and profiled the expression of genes (including *BcCND1*, *BcCND2*, *BcCND4*, *BcCND5*, *BcCND8* and *BcCND10*) related to Ca^2+^/calcineurin-signal [17,37]. Our results indicated that in the presence of CsA, the expression levels of the tested CND genes in the WT strain were downregulated except for that of *BcCND8*, compared to that in the absence of CsA (Figure 7a). Interestingly, in the mutant Δ*Bccyp2*, *BcCND2* was not suppressed by CsA, suggesting that the suppression of this CND gene is in a BcCyp2-dependent fashion. In addition, BcCyp2 is likely involved in the upregulation of *BcCND8* by CsA, since disruption of *BcCYP2* increased the degree of its upregulation (Figure 7b). Our data demonstrate that BcCyp2 is involved in mediating the expression of CND genes regulated by CsA.

### 2.10. BcCyp2 Functions under the Control of Histone 3 Lysine 4 (H3K4) Demethylase BcJar1

H3K4 demethylase BcJar1 regulates *B. cinerea* infection-related development, including IC formation, and virulence via orchestrating genome-wide expression of the pathogenesis-related gene [36]. To determine whether the expression of *BcCYP2* was regulated by H3K4 demethylase BcJar1 during IC formation, we cultured the WT and Δ*Bcjar1* mutant strains and compared the expression level of *BcCYP2* in the tested strains during IC formation. Our result indicated that the expression level of *BcCYP2* in the mutant strain dramatically reduced (Figure 7c), suggesting that BcCyp2 functions under the control of BcJar1 in mediating IC formation.

## 3. Discussion

In this work, we screened and identified a virulence-attenuated mutant from our *B. cinerea* T-DNA insertion library containing ~50,000 transformants [31,32,33]. Bioinformatics analysis indicates that the identified hypothetical protein BcCyp2 is a member of the cyclophilin_ABH_like protein family that contains cyclophilin A, B and H-like cyclophilin-type PPIase domain (Appendix A; Appendix A). The family contains cyclophilin proteins similar to human cyclophilins CypA, CypB (hCyP-19) and CypH implicated in protein folding processes, which depend on their catalytic/chaperone-like activities. Human CypA, CypB, *Saccharomyces cerevisiae* Cpr1 and *Caenorhabditis elegans* Cyp-3, are inhibited by the immunosuppressant CsA (https://www.ncbi.nlm.nih.gov/Structure/cdd/cddsrv.cgi (accessed on 1 January 2021)). Our data of the strong inhibition of hyphal morphogenesis (e.g., hyphal branching, infection structure formation) and virulence of the strains harboring *BcCYP2* by CsA (Figure 6) support the results from bioinformatics analysis.

Cyclophilins play diverse roles in the development and pathogenesis of pathogenic fungal pathogens. In the human fungal pathogen *Cryptococcus neoformans*, cyclophilin A proteins Cpa1 and Cpa2 play an overlapping role in cell growth, mating, virulence and CsA toxicity. Cpa1 and Cpa2 also have divergent functions. The *cpa1* mutants are inviable at high temperatures and virulence-attenuated, whereas the *cpa2* mutants are viable at 39 °C and fully virulent. The *cpa1 cpa2* double mutants are severely virulence-attenuated and inviable at high temperatures. Both Cpa1 and Cpa2 mediate sensitivity to CsA, suggesting that either protein can form a CsA/CypA complex and inactivate calcineurin [38]. MoCyp1 is a virulence-associated factor in *M. oryzae*. The cyclophilin MoCyp1 regulates infection-related functions, e.g., penetration peg formation and appressorium turgor generation, and also is required for efficient conidiation [29]. In the chestnut blight fungus *C. parasitica*, CpCyp1 is a hypovirus-regulated cyclophilin and is a virulence-associated factor. Loss of *CpCYP1* reduces transcript levels for genes encoding key components of the heterotrimeric guanosine triphosphate (GTP)-binding protein (G-protein) signaling pathway that is essential for sensing environmental cues and are involved in *C. parasitica* development and virulence [30]. Cyclophilins in the insect pathogenic fungus *B. bassiana* display common and unique roles in temperature sensitivity, hyphal growth, conidiation, cyclosporine resistance, and virulence [28].

Loss of cyclophilin Bcp1 in *B. cinerea* does not affect appressorium formation, but reduces the virulence of the pathogen [17]. Similar to these findings, we found that disruption of *BcCYP2* did not affect appressorium formation, vegetative growth, and asexual reproduction, but delayed IC formation (Figure 3). The delay of IC formation may be an important reason in virulence reduction of the *BcCYP2* deletion mutants (inefficient penetration is another reason). Many phytopathogenic fungi can recognize hydrophobic surface to initiate appressorium formation via the cAMP signaling pathway, and the differentiation of appressoria and invasive hyphae is mediated by the Pmk1-MAPK cascade [39,40,41,42]. Exogenous cAMP restored the ability of the Δ*Bccyp2* mutant strain to form ICs to the level of the WT and/or complemented strains without supplemented with cAMP, suggesting that loss of cyclophilin BcCyp2 in *B. cinerea* does not impair the cellular response to cAMP during IC formation, and that BcCyp2 is located in the upstream of the cAMP pathway. Interestingly, during IC formation, the expression level of *BcCYP2* was regulated by H3K4 demethylase BcJar1 (Figure 7c), a pathogen factor that regulates fungal infection-related development and virulence via controlling the expression of pathogenesis-related genes [36]. The association of BcJar1 with BcCyp2 in the mediation of IC formation remains to be characterized.

Calmodulin-dependent signaling is involved in infection structure formation in some phytopathogenic fungi. In *M. oryzae*, *CYP1* may encode a cellular target cyclophilin for CsA since the loss of the gene provides high-level resistance to CsA. CsA-mediated calcineurin inhibition greatly affects hyphal development and impairs appressorium formation, indicating that calcineurin is required for appressorium morphogenesis [29]. Our data of the Δ*Bccyp2* mutants displaying less affected hyphal development and infection structure formation in the presence of CsA (Figure 6) support the view. 

CsA (in low concentrations) altered hyphal morphology and inhibited infection structure formation of the strains harboring *BcCYP2*, but less affected these development processes in strains lacking *BcCYP2*, suggesting that *BcCYP2* is a receptor of CsA and their interaction may also inhibit the activity of calcineurin, a crucial regulator of many cellular responses or physiological processes, including hyphal morphology, infection structure formation, stress adaptation, and virulence in *B. cinerea* [17,21]. However, in the presence of higher concentration (>0.4 µg/mL) of CsA, the strain harboring or lacking *BcCYP2* displayed similar levels in hyphal morphological alteration, infection structure formation, and virulence (Figure 6), suggesting that in *B. cinerea*, there may be other cyclophilins serving as CsA receptors. Our identification of 16 cyclophilin-like proteins in *B. cinerea* (Appendix A) and the reported Bcp1 [17,21] supports the view. Among the identified 16 cyclophilin-like proteins, XP_001559584.2 (Bcp1), XP_001559969.1 (BcCyp2), AAQ16573.1, EMR81994.1, XP_001547831.1, XP_001556077.1, XP_001559491.1, and XP_024549591.1 belong to the cyclophilin ABH_like protein family. These putative cyclophilins, including the reported Bcp1 and BcCyp2 described here (Appendix A), may serve as candidate CsA receptors. The observation of morphological alteration in the Δ*Bccyp2* mutants in the presence of a higher concentration of CsA may thus result from functional redundancy in the CsA receptor of other cyclophilins in the gray mold fungus.

Based on our findings, we proposed a model to describe cyclophilin BcCyp2 mediation of infection-related development and virulence in *B. cinerea* (Figure 7d). Several lines of evidence support the working model. First, the cyclophilin BcCyp2 regulates infection-related development (e.g., IC formation) and virulence. These processes may result from the interaction of the cyclophilin with calcineurin, as evidenced by our finding that *BcCYP2* mediates the expression of CND genes in the absence of CsA (Appendix A), which may ultimately alter calcineurin function. Cyclophilins serve as virulence-associated factors and fulfill infection-related functions in many pathogenic fungi [17,28,29,38]. Our data indicate that loss of *BcCYP2* does not impair hyphal development and appressorium formation, indicating that in *B. cinerea*, regulation of hyphal development and appressorium formation are not natural functions of cyclophilin. Second, cyclophilins including BcCyp2 (this study) and Bcp1 [17] are necessary for CsA inhibition of infection-related development and virulence. In the presence of CsA, the formation of a CsA/cyclophilin complex targets calcineurin and prevents its activity in the regulation of fungal morphogenesis, e.g., hyphal branching/development, infection structure formation [11,12,17,21,29]. Our findings on the differential inhibition of hyphal development, infection structure formation, and host infection in the strains harboring or lacking *BcCYP2* in the presence of CsA support this view (Figure 6). Third, cyclophilin BcCyp2 functions in the upstream of cAMP-signaling that regulates the infection-related development of phytopathogenic fungi. Deletion of the cyclophilin in *B. cinerea* results in the delay of IC formation, and exogenous cAMP rescues the defect in IC formation of the mutant strains (Figure 5). Finally, cyclophilin BcCyp2 functions in morphogenesis and virulence may be through direct regulation of CND gene expression. Similar to the previous findings on Bcp1 or CsA mediation of CND gene expression [17,21], in the presence of CsA, most tested CND genes in both the WT and mutant strains were downregulated, except for *CND2*, whose suppression by CsA is specifically dependent on BcCyp2 (Figure 7a,b). These findings suggest that BcCyp2 functions in the upstream of Ca^2+^/calmodulin-dependent phosphatase calcineurin signaling.

Collectively, we identified BcCyp2 as a functional cyclophilin in *B. cinerea* and demonstrated that BcCyp2 is a virulence-associated factor that functions in fungal morphogenesis and virulence under the control of H3K4 demethylase Jar1 as well as in the upstream of cAMP- and Ca^2+^/calmodulin-dependent phosphatase calcineurin signaling pathways. CsA reduced virulence of the *BcCYP2*-harboring strains via impairment of their infection structure formation and host penetration. Our work reveals fundamental roles for cyclophilins in phytopathogenic fungi and provides new insights into the pathogenesis of *B. cinerea*.

## 4. Materials and Methods

### 4.1. Fungal Strains and Culture Conditions

*B. cinerea* WT strain B05.10 and its derived strains, including mutants Δ*Bccyp2*-1, Δ*Bccyp2*-2, and the Δ*Bccyp2*-1 complemented strain Δ*Bccyp2*-C, were cultured on potato dextrose agar (PDA) medium or CM plates as the methods previously described [6,31,32,33].

### 4.2. Identification of BcCYP2 as a Pathogenicity-Related Gene

A virulence-attenuated mutant strain M331 was identified from a *B. cinerea* T-DNA insertion library containing ~50,000 transformants via screening for pathogenicity-attenuated mutants using detached tomato and strawberry leaves as hosts [31,32,33]. The thermal asymmetric interlaced PCR approach [31,32,33] was used to analyze T-DNA insertion region in the genome of M331 with the primers listed in Appendix A. T-DNA flanking sequences were identified by using BlastN analysis in *B. cinerea* database (http://fungi.ensembl.org/Botrytis_cinerea/Info/Index (accessed on 1 January 2021)). Sequence alignment of homologous proteins was performed by using GENEDOC (http://nrbsc.org/gfx/genedoc (accessed on 1 January 2021)).

### 4.3. Generation of Gene Deletion and the Corresponding Genetic Complemented Strains

The generation of *BcCYP2* deletion mutants was performed using the gene replacement method described previously [6,31,32,33,43]. The vector pXEH containing the hygromycin phosphotransferase gene (*HPH*) was used to replace the target gene [6,31,32,36]. The 5′- and 3′- homologous flanks of the target gene were amplified and cloned into pXEH in the upstream and downstream of *HPH*, respectively. The resultant KO vector was transformed into *A. tumefaciens* strain AGL-1 as previously described [32]. The ATMT method was used to obtain fungal transformants [44]. The gene deleted transformants of *B. cinerea* were screened on PDA supplemented with 100 μg/mL hygromycin.

The vector pSULPH, resistant to chlorimuron ethyl, was used to generate the complemented strain Δ*Bccyp2*-C [6,35,36]. The fragment containing 1156 bp upstream and 1070 bp downstream of *BcCYP2* coding region was amplified by PCR and cloned into pSULPH to generate the complementary vector. The complementary vector was transformed into *A. tumefaciens* strain AGL-1 via the ATMT approach. The resultant transformants of *B. cinerea* were screened on a defined complex medium (DCM) containing 100 μg/mL chlorimuron ethyl.

The transformants were screened by PCR amplification. The *BcCYP2* gene deletion mutants and complemented strains were further confirmed by qRT-PCR [2]. The primers used in these experiments are listed in Appendix A. DNA and RNA were extracted as previously described [45,46].

### 4.4. Fungal Developmental Assays

The growth of the tested *B. cinerea* strains was determined by measuring the radial diameter of colonies on CM as previously described [6,31,32,33,36]. Fresh conidia of the WT, Δ*Bccyp2* and Δ*Bccyp2*-C strains were harvested from PDA or CM plates. For the measurement of conidial morphology and germination, the concentration of the conidial suspension was adjusted to 1 × 10^6^ conidia/mL. For determination of IC formation, liquid CM droplets (18 μL) were placed on the surface of glass slides, then conidial suspension (1 × 10^6^ conidia/mL, 2 μL) was added and quickly mixed. The inoculated slides were then incubated in a wetting chamber at 20 °C. At 24 hpi, formation of IC was observed and imaged with a microscope (Nikon Eclipse 80i Fluorescence Microscope System, Nikon, Japan). For each sample, IC formation was quantitatively determined via a comparison of the total number of ICs from five randomly selected fields of view under the microscope. Conidial germination, appressorium formation and other phenotypic assays were also analyses with the microscopy system. The experiments were repeated at least three times.

### 4.5. Plant Infection and Cytological Assays

Conidia were harvested from 10-day old cultures on PDA at 20 °C. Conidial droplets (2 × 10^5^ conidia/mL in ½ liquid CM, 5 µL) of the WT, Δ*Bccyp2* and Δ*Bccyp2*-C strains were inoculated on green bean leaves [32,33,35,36]. For pathogenicity assays using mycelia, mycelial plugs (5 mm in diameter) from 3-day old culture of the tested strains were inoculated on surfaces of the related plant materials, including green bean leaves and apple fruits. All the inoculated materials were incubated in plastic containers with a sheet of plastic film sealed on the top of each container to maintain a high humidity of infection conditions. *B. cinerea* inoculated leaves (at 0, 48, 60, and 72 hpi for inoculation with conidial suspension; at 24, 48, and 72 hpi for inoculation with mycelial plugs), apple fruits (at 24, 48 and 72 hpi for inoculation with mycelial plugs) were photographically documented at the indicated time points post-inoculation in the parentheses. Quantification of lesion size was analyzed using ImageJ software (https://imagej.nih.gov/ij/download.html (accessed on 1 January 2021)).

Hydrophobic sides of onion epidermal cells were inoculated with conidial suspension droplets (2 × 10^5^ conidia/mL, 5 µL) and cultured at 20 °C in a humid environment. At 12 and 24 hpi, the inoculated epidermal cells were stained with lactophenol aniline blue for host penetration assay with the microscopy system.

### 4.6. Stress Adaptation Assays

The sensitivity of the test strains to different stresses was analyzed as previously described [6,31,32,36]. Conidial droplets (1 × 10^6^ conidia/mL, 1 μL) or mycelial plugs (5 mm in diameter) of the WT, *BcCYP2* deletion mutant and complemented strains were inoculated on CM supplemented with different stress agents including osmotic stress agents NaCl and KCl (1 M for each), oxidative stress agent H_2_O_2_ (5 mM), cell wall disturbing agents SDS (0.005%) and Congo Red (300 μg/mL). All the inoculated plates were incubated at 20 °C in the dark, and the adaptabilities of the strains to these stress agents were evaluated by comparison of their colony diameters. At least three independent experiments were performed, and triple plates of each strain were measured in each test.

### 4.7. CsA Sensitivity Assay

CsA was dissolved in dimethyl sulfoxide (DMSO). Liquid CM containing CsA was mixed with conidial suspension (1 × 10^5^ conidia/mL, vol:vol = 1:1) to make the final CsA concentration of 0.2, 0.4, 0.6, 0.8 and 1.0 μg/mL, respectively, and 200 μL of the mixture was placed in 96-well plates and cultured at 20 °C in the dark. At 24 or 48 hpi, conidial germination and mycelial morphology were observed and imaged using the microscopy system. For the effects of CsA on infection structure formation and host penetration of the tested strains, the droplets (5 × 10^4^ conidia/mL, 20 μL) of the mixture of 0.4 μg/mL CsA with the conidia suspension were inoculated on onion epidermis and incubated at 20 °C in the dark. The inoculation treated with DMSO was used as a control, since CsA was dissolved in DMSO. At 24 or 48 hpi, the appressorium formation and host penetration were determined and compared. All the CsA inhibition experiments were repeated three times.

### 4.8. cAMP Sensitivity Assay

The effect of cAMP on infection structure formation of the WT, Δ*Bccyp2*, and complemented strains was evaluated as we previously described [6]. Briefly, conidial suspensions of the indicated strains, supplemented with cAMP at the concentration of 0, 20, 50, or 80 μM, were inoculated on glass slides and incubated in the dark for 12, 18, 24 or 36 hpi. At the indicated hpi, the size of ICs was determined and compared.

### 4.9. Gene Expression Profiling Assays

Conidia of the WT and Δ*Bccyp2* strains were grown in liquid YG medium [yeast extract, 2 g/L; glucose,10 g/L; KH_2_PO_4_, 2 g/L; K_2_HPO_4_, 1.5 g/L; (NH_4_)_2_SO_4_, 1 g/L; MgSO_4_·7H_2_O, 0.5 g/L] for 60 h and then moved to the same medium without the nitrogen source. After 4 h of cultivation, CsA (dissolved in 1% DMSO and 99% absolute ethanol) was added to make the final concentration of 10 μg/mL. Following treatment with CsA for 3 h, the mycelia of the tested strains were collected and subjected to RNA extraction using the previously described method [45,46]. The mRNA levels of Ca^2+^/calmodulin signaling genes were determined by the previously described qRT-PCR approach [31] with the primers listed in Appendix A. 

### 4.10. Statistical Analysis

All quantitative data in this study are derived from the results of at least three independent experiments with triplicate samples examined for each strain in each experiment. The data of the control including mycelial growth, lesion size, conidia germination, or other tested phenotypes were standardized to 100% to make the results from different independent experiments comparable. The Student’s *t*-test was used to analyze the data between control (or WT) and treatment (or mutant) groups and *p* < 0.05 was considered a significant difference.

## Figures and Tables

**Figure 1 ijms-22-01694-f001:**
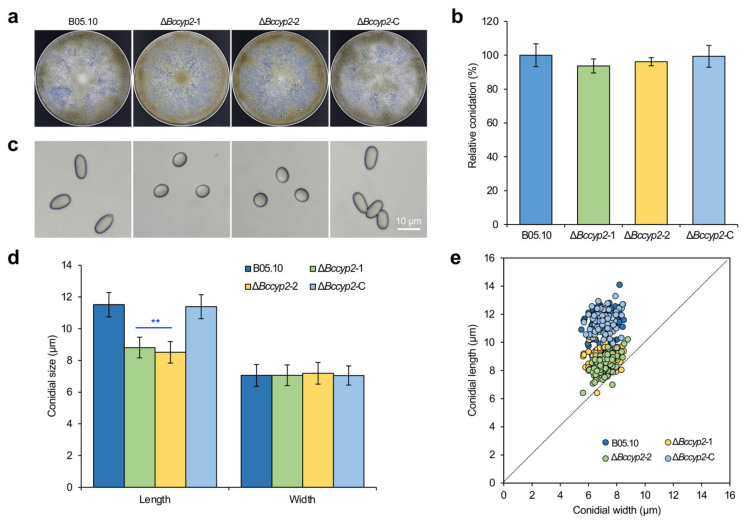
*BcCYP2* is required for *B. cinerea* conidial morphogenesis but dispensable for conidiation. (**a**) Conidiation of the indicated wild-type (WT), Δ*Bccyp2* and complemented strains of *B. cinerea* on complete medium (CM) plates at 10 days post-inoculation/incubation (dpi). (**b**) Quantification of the relative conidiation of the indicated strains shown in (**a**). (**c**) Alteration of conidial morphology in the Δ*Bccyp2* mutants. (**d**) Quantification of sizes (length and width) of conidia produced by the indicated strains. Conidia were harvested from 10-day old CM plates and conidia (>300) of each strain were measured under a microscope. (**e**) Loss of *BcCYP2* increases the number of globose and less elliptical conidia (closer to the dash line). Conidia (>300) of each strain were measured under a microscope in each independent experiment. The representative images are from one of the experiments, three independent experiments were performed and resulted in similar results. Data represent the means ± standard deviations (SD) from three independent experiments with triplicate plates examined for each treatment. **: significance at *p* < 0.01.

**Figure 2 ijms-22-01694-f002:**
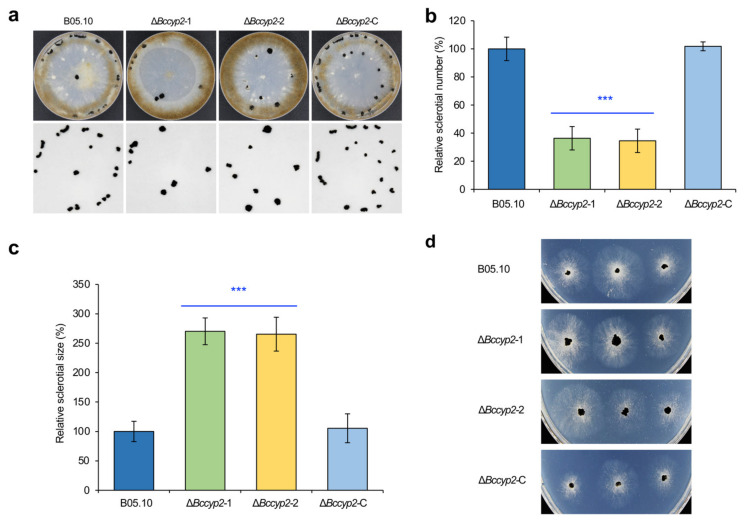
*BcCYP2* mediates *B. cinerea* sclerotium production and morphogenesis. (**a**) Deletion of *BcCYP2* in *B.cinerea* reduces sclerotium production. Conidia of the wild-type (WT), Δ*Bccyp2*, and complemented strains were inoculated on CM plates and cultured at 20 °C in the dark. Sclerotium production was photographically documented at four weeks post-incubation. (**b**) Quantification of sclerotium production. (**c**) Quantitative analysis of sclerotial sizes of the indicated strains via ImageJ software (https://imagej.nih.gov/ij/download.html (accessed on 1 January 2021)). (**d**) Germination of sclerotia produced by the tested strains. The representative images are from one of the experiments, at least three independent experiments were performed, and all the experiments resulted in similar results. Data represent means ± SD from at least three independent experiments. ***: significance at *p* < 0.001.

**Figure 3 ijms-22-01694-f003:**
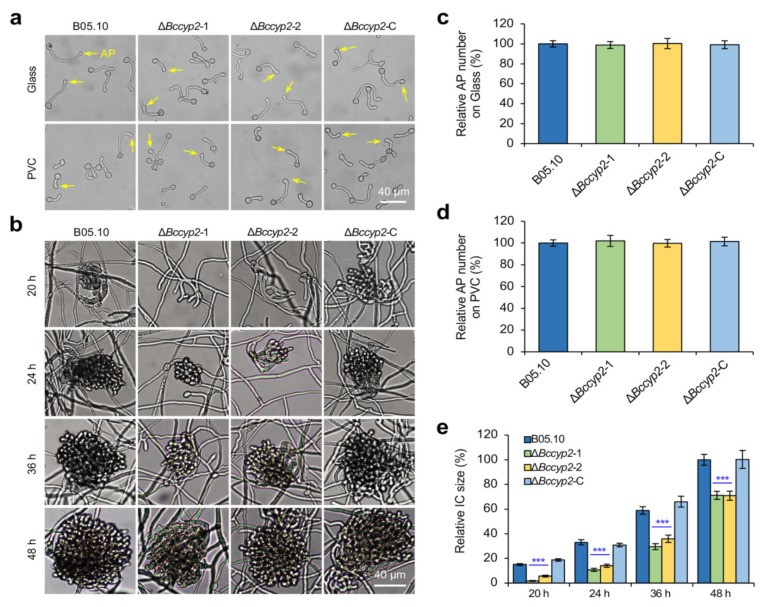
*BcCYP2* plays an important role in infection cushion (IC) formation. (**a**) *BcCYP2* is dispensable for *B. cinerea* appressorium formation. Conidia (5 × 10^5^ conidia/mL in ½ liquid CM, 20 µL) of the indicated strains were inoculated on glass slides or PVC sheets and cultured at 20 °C. At 7 h post-inoculation/incubation (hpi), appressorium formation by the tested strains was observed and documented. (**b**) Loss of *BcCYP2* delays the development of ICs. Conidia (1 × 10^5^ conidia/mL in ½ liquid CM, 20 µL) of the indicated strains were inoculated on glass slides and cultured at 20 °C. ICs were photographically documented at 20, 24, 36, and 48 hpi. Images from a single representative experiment are presented. (**c**,**d**) Quantitative analysis of the relative number of appressoria produced by the tested strains on glass slides (**c**) or PVC sheets (**d**). (**e**) Quantification of the sizes of ICs produced by the tested strains. Data represent means ± SD from three independent experiments in which triplicate slides were examined for each strain in each experiment. ***: significance at *p* < 0.001. AP: Appressorium; PVC: Polyvinyl chloride sheet.

**Figure 4 ijms-22-01694-f004:**
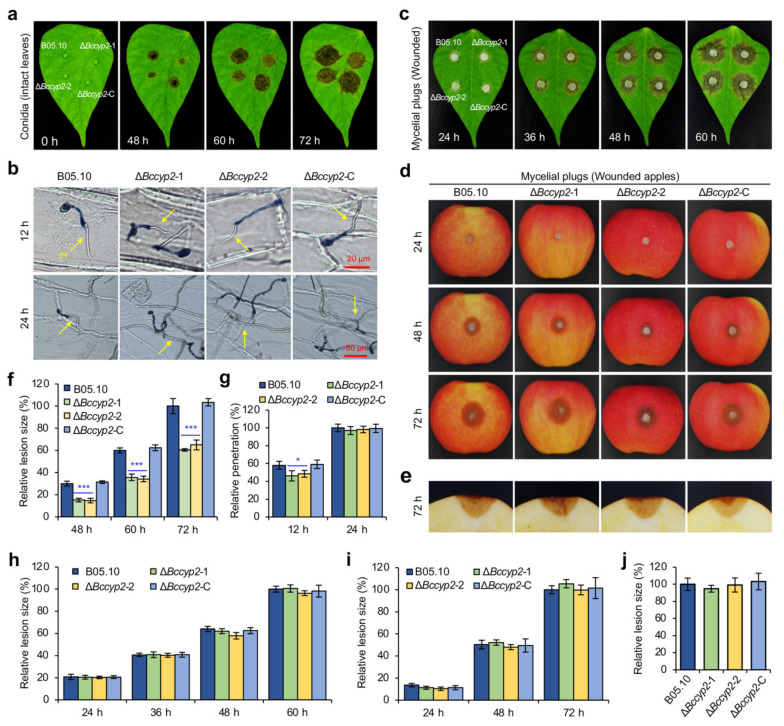
Loss of *BcCYP2* in *B. cinerea* impairs virulence of the pathogen. (**a**) Deletion of *BcCYP2* reduces the virulence of *B. cinerea*. Conidial suspensions (2 × 10^5^ conidia/mL in ½ liquid CM, 5 µL) of the WT, Δ*Bccyp2*, and complement strains were inoculated on intact green bean leaves. At 0, 48, 60, and 72 hpi, the inoculated leaves were photographically documented. (**b**) Penetration assay of the test strains. Conidia (2 × 10^5^ conidia/mL, 5 μL) of the strains were inoculated on onion epidermis and cultured at 20 °C in the dark. At 12 and 24 hpi, the inoculated epidermal cells were stained with lactophenol aniline blue and then observed and photographically documented. (**c**,**d**) Deletion of *BcCYP2* does not affect invasive hyphal expansion in planta. Mycelial plugs of each strain were inoculated on artificially wounded green bean leaves (**c**) or apple fruits (**d**) and the lesions were photographically documented at the indicated hpi. (**e**) The cross-section of lesions on wounded apple fruits. (**f**) Quantification of lesion sizes caused by the indicated strains on green bean leaves shown in (**a**). (**g**) Quantification of penetration by the indicated strains on onion epidermal cells shown in (**b**). (**h**–**i**) Quantification of lesion sizes caused by the indicated strains on wounded green bean leaves or apple fruit shown in (**c**,**d**), respectively. (**j**) Quantification of lesion sizes caused by the indicated strains on apple fruits shown in (**e**). Data represent means ± SD from three independent experiments. *, ***: significance at *p* < 0.05 and 0.001, respectively. IH: invasive hyphae.

**Figure 5 ijms-22-01694-f005:**
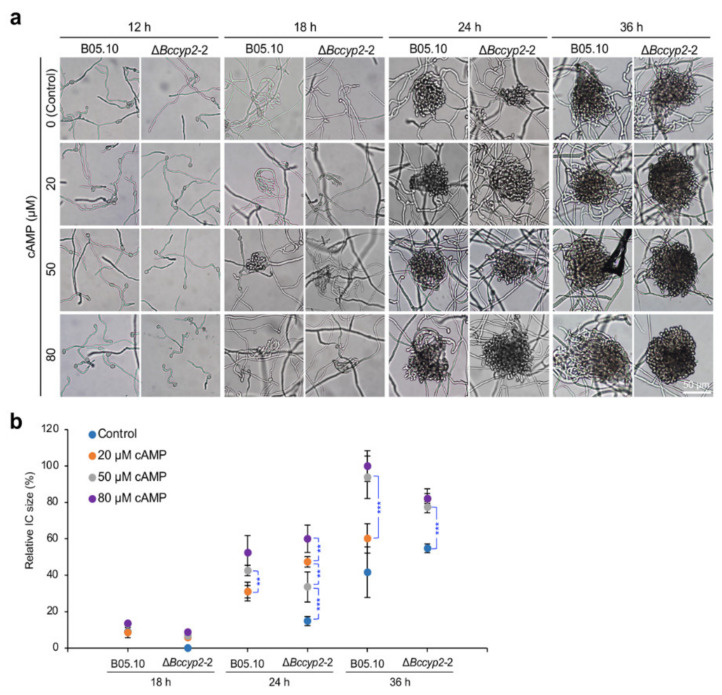
Cyclic adenosine monophosphate (cAMP) restores IC formation of the *BcCYP2* deletion mutants. (**a**) Exogenous cAMP rescues defects of IC formation of the Δ*Bccyp2* mutants. Conidial droplets (10 µL) supplemented with cAMP (0, 20, 50, 80 µM) were inoculated on glass slides and incubated in the dark for 12, 18, 24 or 36 h; the development of ICs was then photographically documented. (**b**) Quantitative analysis of the sizes of ICs produced by the tested strains. IC: Infection cushion. **, ***: significance at *p* < 0.01 and 0.001, respectively.

**Figure 6 ijms-22-01694-f006:**
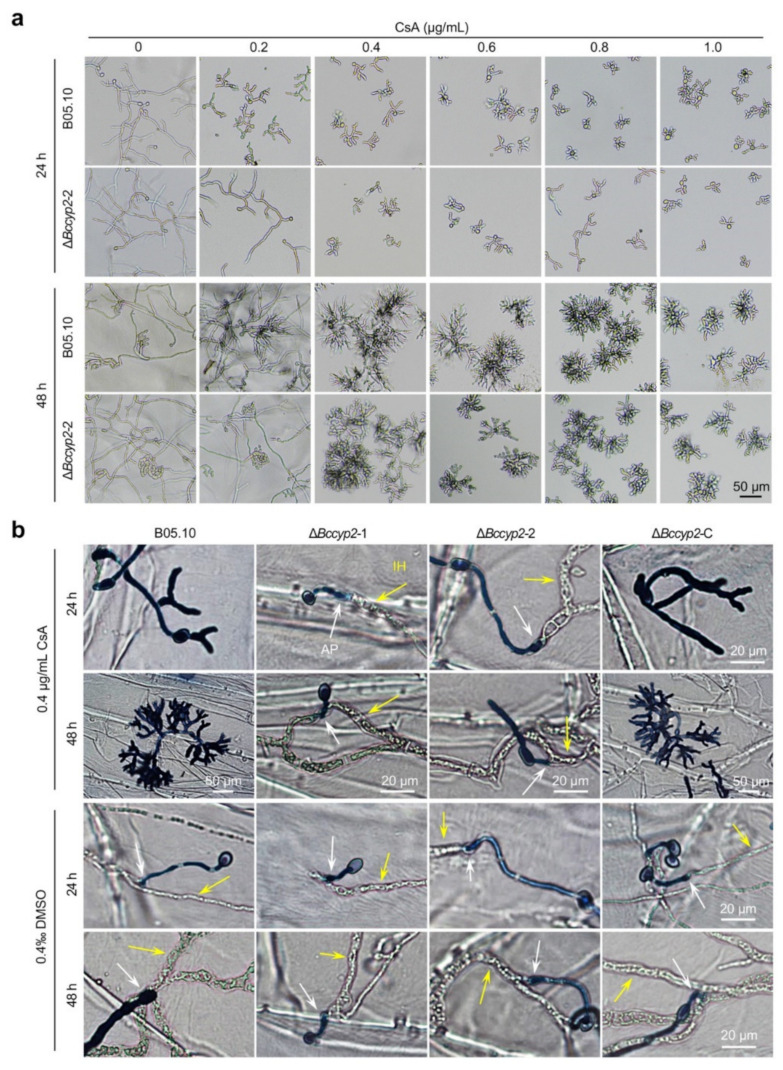
Effects of CsA on infection-related morphogenesis as well as host penetration and growth of invasive hyphae. (**a**) Suppression of *B. cinerea* mycelial growth by CsA. Conidia (5 × 10^4^ conidia/mL in ½ liquid CM, 200 µL) of the indicated strains mixed with CsA (0, 0.2, 0.4, 0.6, 0.8, or 1.0 µg/mL) were inoculated in 96-well plates and cultured at 20 °C. Mycelial morphogenesis of the tested strains was observed and photographically documented at 24 and 48 hpi. (**b**) Suppression of *B. cinerea* infection by CsA. Conidia of each strain (5 × 10^4^ conidia/mL, 20 µL) mixed with or without CsA were inoculated on onion epidermis and cultured at 20 °C. The solvent (DMSO) treated inoculation was used as a control. At 24 or 48 hpi, the inoculated epidermal cells were performed lactophenol aniline blue staining, microscopy observation and imaging. AP: appressorium, IH: invasive hyphae. Representative images are from one of the experiments, at least three independent experiments were performed, and all the experiments resulted in similar results.

**Figure 7 ijms-22-01694-f007:**
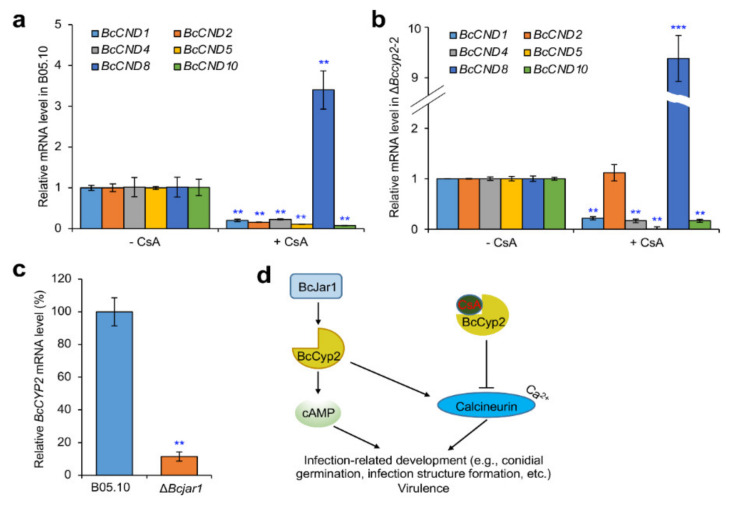
BcCyp2 regulates expression of calcineurin-dependent (CND) genes and is regulated by H3K4 demethylase BcJar1. (**a**,**b**) Relative transcriptional level of CND genes in the WT B05.10 (**a**) and mutant Δ*Bccyp2*-2 (**b**) strains. Conidia of the two strains were grown in liquid YG medium for 60 h and then moved to the same medium without the nitrogen source (YG-N) and incubated for 4 h, the mycelia of the tested strains were further incubated with YG-N medium containing 10 μg/mL CsA for 3 h and then collected for RNA extraction. The mRNA levels of Ca^2+^/calmodulin signaling genes were determined by quantitative reverse transcription PCR (qRT-PCR). (**c**) Loss of *BcJAR1* [36] downregulates *BcCYP2* expression level. Mycelial plugs of B05.10 or Δ*Bcjar1* strain were induced for IC formation. At 36 hpi (during IC formation), mycelia of the tested strains were then collected and subjected to RNA extraction. The mRNA levels of *BcCYP2* were determined by qRT-PCR. (**d**) Proposed models describing BcCyp2 regulation of fungal infection-related development and virulence as well as CsA-mediated functional blocking. Data represent means ± SD from three independent experiments. **, ***: significance at *p* < 0.01 and 0.001, respectively.

## Data Availability

Data is contained within the article and Appendix A.

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
