# Peer review of "Cyclophilin BcCyp2 Regulates Infection-Related Development to Facilitate Virulence of the Gray Mold Fungus Botrytis cinerea"

_ijms, 2021, doi:10.3390/ijms22041694_

Round 1

Reviewer 1 Report

In my opinion, the work entitled "Cyclophilin BcCyp2 regulates infection-related development to facilitate virulence of the gray mold fungus Botrytis cinerea" is of high interest and deserves to be published in International Journal Molecular Science. However, some minor errors must be previously corrected prior to publication.

Minor revision

In page 2 line 79 change environmental stresses for environmental stress.

In page 3 line 121 change (Fig-ure S1b) for (Figure S1b). Delete the dash.

in the legend of figure 4 you must correct ml for mL and μl for μL. Please review the entire text and homogenize the terminology throughout it.

Author Response

Point 1: In page 2 line 79 change environmental stresses for environmental stress.

Response: “environmental stresses” has been replaced by “environmental stress” in our revised manuscript (Page 2 line 76) as per the suggestion of the reviewer.

Point 2: In page 3 line 121 change (Fig-ure S1b) for (Figure S1b). Delete the dash.

Response: “(Fig-ure S1b)” has been replaced by “(Figure S1b)” in our revised manuscript (Page 3 line121).

Point 3: in the legend of figure 4 you must correct ml for mL and μl for μL. Please review the entire text and homogenize the terminology throughout it.

Response: The “ml” and “μl” have been corrected as “mL” and “μL”, respectively, throughout the manuscript, including in Figure 6 and the Supplemental Information.

Reviewer 2 Report

The manuscript ijms-1084125 describes the involvement of cyclophilin BcCyp2 in fungal morphogenesis and pathogenicity of Botrytis cinerea. These results provide useful information for developing effective fungicides for controlling B. cinerea. However, there are some problems in this manuscript that needed to be addressed.

Line 146. “or 15” should be deleted.

Line 470, 476. The transformation method of B. cinerea should be described.

Line 488, 525. The conidia concentration is inconsistent with that of the figure legend.

Line 549. The method of qRT-PCR should be described in detail.

Author Response

Point 1: Line 146. “or 15” should be deleted.

Response: “or 15” has been removed in our revised manuscript (Page 3 line145).

Point 2: Line 470, 476. The transformation method of B. cinerea should be described.

Response: This is an excellent comment. We did not clearly mention the fungal transformation method in our previous submission. The transformation method of B. cinerea, which was previously described in detail in the cited references, has been included in our revised manuscript (Page 14 lines 493, 494 and 500, or Page 14 lines 464, 465 and 471 in the submission version with or without track changes, respectively).

Point 3: Line 488, 525. The conidia concentration is inconsistent with that of the figure legend.

Response: We appreciate the reviewer pointing out this issue. The conidial concentrations in the Method section and those of the main text and figure legend have been verified in the revised manuscript throughout the manuscript, including in the Supplemental Information.

Point 4: Line 549. The method of qRT-PCR should be described in detail.

Response: The detailed method of qRT-PCR was previously described in the cited references in our revised manuscript (Page 15 line 585 or Page 15 lines 545 in the submission version with or without track changes).
